# Understanding quantum machine learning also requires rethinking generalization

Elies Gil-Fuster [1,2], Jens Eisert [1,2,3] & Carlos Bravo-Prieto [1]

Quantum machine learning models have shown successful generalization performance even when trained with few data. In this work, through systematic randomization experiments, we show that traditional approaches to understanding generalization fail to explain the behavior of such quantum models. Our experiments reveal that state-of-the-art quantum neural networks accurately fit random states and random labeling of training data. This ability to memorize random data defies current notions of small generalization error, problematizing approaches that build on complexity measures such as the VC dimension, the Rademacher complexity, and all their uniform relatives. We complement our empirical results with a theoretical construction showing that quantum neural networks can fit arbitrary labels to quantum states, hinting at their memorization ability. Our results do not preclude the possibility of good generalization with few training data but rather rule out any possible guarantees based only on the properties of the model family. These findings expose a fundamental challenge in the conventional understanding of generalization in quantum machine learning and highlight the need for a paradigm shift in the study of quantum models for machine learning tasks.

Quantum devices promise applications in solving computational problems beyond the capabilities of classical computers[1–5]. Given the paramount importance of machine learning in a wide variety of algorithmic applications that make predictions based on training data, it is a natural thought to investigate to what extent quantum computers may assist in tackling machine learning tasks. Indeed, such tasks are commonly listed among the most promising candidate applications for near-term quantum devices[6–9]. To date, within this emergent field of quantum machine learning (QML) a body of literature is available that heuristically explores the potential of improving learning algorithms by having access to quantum devices[10–20]. Among the models considered, parameterized quantum circuits (PQCs), also known as quantum neural networks (QNNs), take center stage in those considerations[21–23]. For fine-tuned problems in quantum machine learning, quantum advantages in computational complexity have been proven over classical computers[24–27], but to date, such advantages rely on the availability of full-scale quantum computers, not being within reach for near-term

architectures. While for PQCs such an advantage has not been shown yet, a growing body of literature is available that investigates their expressivity[28–34], trainability[35–44], and generalization[45–60]—basically aimed at understanding what to expect from such quantum models. Among those studies, the latter notions of generalization are particularly important since they are aimed at providing guarantees on the performance of QML models with unseen data after the training process.

The importance of notions of generalization for PQCs is actually reflecting the development in classical machine learning: Vapnik's contributions[61] have laid the groundwork for the formal study of statistical learning systems. This methodology was considered standard in classical machine learning theory until roughly the last decade. However, the mindset put forth in this work has been disrupted by seminal work[62] demonstrating that the conventional understanding of generalization is unable to explain the great success of large-scale deep convolutional neural networks. These networks, which display orders of magnitude more trainable parameters than the dimensions of the

[1]Dahlem Center for Complex Quantum Systems, Freie Universität Berlin, Berlin, Germany. [2]Fraunhofer Heinrich Hertz Institute, Berlin, Germany. [3]Helmholtz-Zentrum Berlin für Materialien und Energie, Berlin, Germany. ✉e-mail: jense@zedat.fu-berlin.de; c.bravo.prieto@fu-berlin.de

images they process, defied conventional wisdom concerning generalization.

Employing clever randomization tests derived from non-parametric statistics[63], the authors of ref. 62 exposed cracks in the foundations of Vapnik's theory and its successors[64], at least when applied to specific, state-of-the-art, large networks. Established complexity measures, such as the well-known VC dimension or Rademacher complexity[65], among others, were inadequate in explaining the generalization behavior of large classical neural networks. Their findings, in the form of numerical experiments, directly challenge many of the well-established uniform generalization bounds for learning models, such as those derived in, e.g., refs. 66–68. Uniform generalization bounds apply uniformly to all hypotheses across an entire function family. Consequently, they fail to distinguish between hypotheses with good out-of-sample performance and those which completely overfit the training data. Moreover, uniform generalization bounds are oblivious to the difference between real-world data and randomly corrupted patterns. This inherent uniformity is what grants long reach to the randomization tests: exposing a single instance of poor generalization is sufficient to reduce the statements of mathematical theorems to mere trivially loose bounds.

This state of affairs has important consequences for the emergent field of QML, as we explore here. Noteworthy, current generalization bounds in quantum machine learning models have essentially uniquely focused on uniform variants. Consequently, our present comprehension remains akin to the classical machine learning canon before the advent of ref. 62. This observation raises a natural question as to whether the same randomization tests would yield analogous outcomes when applied to quantum models. In classical machine learning, it is widely acknowledged that the scale of deep neural networks plays a crucial role in generalization. Analogously, it is widely accepted that current QML models are considerably distant from that size scale. In this context, one would not anticipate similarities between current QML models and high-achieving classical learning models[56,57].

In this article, we provide empirical, long-reaching evidence of unexpected behavior in the field of generalization, with quite arresting conclusions. In fact, we are in the position to challenge notions of generalization, building on similar randomization tests that have been used in ref. 62. As it turns out, they already yield surprising results when applied to near-term QML models employing quantum states as inputs. Our empirical findings, also in the form of numerical experiments, reveal that uniform generalization bounds may not be the right approach for current-scale QML. To corroborate this body of numerical work with a rigorous underpinning, we show how QML models can assign arbitrary labels to quantum states. Specifically, we show that PQCs are able to perfectly fit training sets of polynomial size in the number of qubits. By revealing this ability to memorize random data, our results rule out the good generalization guarantees with few training data from uniform bounds[54,58]. To clarify, our experiments do not study the generalization capacity of state-of-the-art QML. Instead, we expose the limitation of uniform generalization bounds when applied to these models. While QML models have demonstrated good generalization performance in some settings[20,47,54,58,69–71], our contributions do not explain why or how they achieve it. We highlight that the reasons behind their successful generalization remain elusive.

## Results
### Statistical learning theory background
We begin by briefly introducing the necessary terminology for discussing our findings in the framework of supervised learning. We denote $\mathcal{X}$ as the input domain and $\mathcal{Y}$ as the set of possible labels. We assume there is an unknown but fixed distribution $\mathcal{D}(\mathcal{X} \times \mathcal{Y})$ from which the data originate. Let $\mathcal{F}$ represent the family of functions that map $\mathcal{X}$ to $\mathcal{Y}$. The expected risk functional $R$ then quantifies the predictive accuracy of a given function $f$ for data sampled according to $\mathcal{D}$.

The training set, denoted as $S$, comprises $N$ samples drawn from $\mathcal{D}$. The empirical risk $\hat{R}_S(f)$ then evaluates the performance of a function $f$ on the restricted set $S$. The difference between $R(f)$ and $\hat{R}_S(f)$ is referred to as the generalization gap, defined as

$$\text{gen}(f) := |R(f) - \hat{R}_S(f)|. \tag{1}$$

The dependence of gen($f$) on $S$ is implied, as evident from the context. Similarly, the dependence of $R(f)$, $\hat{R}_S(f)$, and gen($f$) on $\mathcal{D}$ is also implicit. We employ $C(\mathcal{F})$ to represent any complexity measure of a function family, such as the VC dimension, the Rademacher complexity, or others[65]. It is important to note that these measures are properties of the whole function family $\mathcal{F}$, and not of single functions $f \in \mathcal{F}$.

In the traditional framework of statistical learning, the way in which the aforementioned concepts relate to one another is as follows. The primary goal of supervised learning is to minimize the expected risk $R$ associated to a learning task, which is an unattainable goal by construction. The so-called bias-variance trade-off stems from rewriting the expected risk as a sum of the two terms

$$R(f) = \underbrace{\hat{R}_S(f)}_{\text{Empirical risk, bias}} + \underbrace{R(f) - \hat{R}_S(f)}_{\text{Generalization gap, variance}}. \tag{2}$$

This characterization as a trade-off arises from the conventional understanding that diminishing one of these components invariably leads to an increase of the other. Two negative scenarios exist at the extremes of the trade-off. Underfitting occurs when the model exhibits high bias, resulting in an imperfect classification of the training set. Conversely, overfitting arises when the model displays high variance, leading to a perfect classification of the training set. Overfitting is considered detrimental as it may cause the learning models to learn spurious correlations induced by noise in the training data. Accommodating this noise in the data would consequently lead to suboptimal performance on new data, i.e., poor generalization. Concerning the model selection problem, practitioners are thus tasked with identifying a model with the appropriate model capacity for each learning task, aiming to strike a balance in the trade-off. These notions are explained more extensively in refs. 52,53.

The previously described scenario is no longer applicable, as demonstrated below. Modern-day (quantum) learning models display good generalization performance while being able to completely overfit the data. This phenomenon is sometimes linked to the ability of learning models to memorize data. The term memorization is defined here as the occurrence of overfitting without concurrent generalization. It is essential to clarify that overfitting, in this context, means perfect fitting of the training set, regardless of its generalization performance. Furthermore, a model is considered to have memorized a training set when both overfitting and poor generalization occur simultaneously. Overall, a high model capacity, particularly in relation to memorization ability, is found to be non-detrimental in addressing learning tasks of practical significance. This phenomenon was initially characterized for large (overparameterized) deep neural networks in ref. 62. In this manuscript, we present analogous, unexpected behavior for current-scale (non-overparameterized) parameterized quantum circuits.

### Randomization tests
Our goal is to improve our understanding of PQCs as learning models. In particular, we tread in the domain of generalization and its interplay with the ability to memorize random data. The main idea of our work builds on the theory of randomization tests from non-parametric statistics[63]. Figure 1 contains a visualization of our framework.

Initially, we train QNNs on quantum states whose labels have been randomized and compare the training accuracy achieved by the same learning model when trained on the true labels. Our results reveal that,

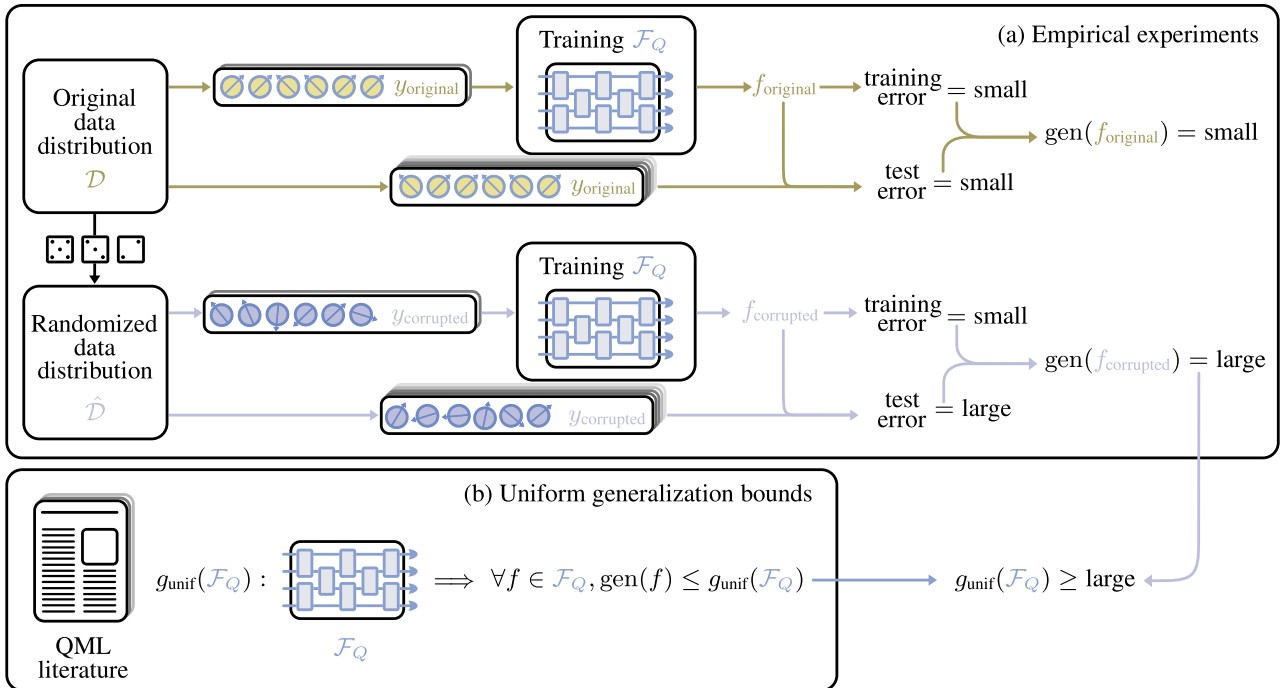

**Fig. 1 | Visualization of our framework. a** In the empirical experiments, a distribution of labeled quantum data $\mathcal{D}$ undergoes a randomization process, leading to a corrupted data distribution $\hat{\mathcal{D}}$. The training and a test set are drawn independently from each distribution. Then, the training sets are fed into an optimization algorithm, which is employed to identify the best fit for each data set individually from a family of parameterized quantum circuits $\mathcal{F}_Q$. This process generates two hypotheses: one for the original data $f_{\text{original}}$ and another for the corrupted data $f_{\text{corrupted}}$. We empirically find that the labeling functions can perfectly fit the training data, leading to small training errors. In parallel, $f_{\text{original}}$ achieves a small test error, indicating good learning performance, and quantified by a small generalization gap $\text{gen}(f_{\text{original}}) = \text{small}$. On the contrary, the randomization process causes $f_{\text{corrupted}}$ to achieve a large test error, which in turn results in a large generalization gap

$\text{gen}(f_{\text{corrupted}}) = \text{large}$. **b** Regarding uniform generalization bounds, it is worth noting that this corner of QML literature assigns the same upper bound $g_{\text{unif}}$ to the entire function family without considering the specific characteristics of each individual function. Finally, we combine two significant findings: (1) We have identified a hypothesis with a large empirical generalization gap, and (2) the uniform generalization bounds impose identical upper bounds on all hypotheses. Consequently, we conclude that any uniform generalization bound derived from the literature must be regarded as "large", indicating that all such bounds are loose for that training data size. The notion of loose generalization bound does not exclude the possibility of achieving good generalization; rather, it fails to explain or predict such successful behavior.

in many cases, the models learn to classify the training data perfectly, regardless of whether the labels have been randomized. By altering the input data, we reach our first finding:

**Observation 1**. (Fitting random labels). Existing QML models can accurately fit random labels to quantum states.

Next, we randomize only a fraction of the labels. We observe a steady increase in the generalization error as the label noise rises. This suggests that QNNs are capable of extracting the residual signal in the data while simultaneously fitting the noisy portion using brute-force memorization.

**Observation 2**. (Fitting partially corrupted labels). Existing QML models can accurately fit partially corrupted labels to quantum states.

In addition to randomizing the labels, we also explore the effects of randomizing the input quantum states themselves and conclude:

**Observation 3**. (Fitting random quantum states). Existing QML models can accurately fit labels to random quantum states.

These randomization experiments result in a remarkably large generalization gap after training without changing the circuit structure, the number of parameters, the number of training examples, or the learning algorithm. As highlighted in ref. 62 for classical learning models, these straightforward experiments have far-reaching implications:

1. Quantum neural networks already show memorization capability for quantum data.

2. The trainability of a model remains largely unaffected by the absence of correlation between input states and labels.
3. Randomizing the labels does not change any properties of the learning task other than the data itself.

In the following, we present our experimental design and the formal interpretation of our results. Even though it would seem that our results contradict established theorems, we elucidate how and why we can prove that uniform generalization bounds are vacuous for currently tested models.

## Numerical results
Here, we show the numerical results of our randomization tests, focusing on a candidate architecture and a well-established classification problem: the quantum convolutional neural network (QCNN)[69] and the classification of quantum phases of matter.

Classifying quantum phases of matter accurately is a relevant task for the study of condensed-matter physics[72,73]. Moreover, due to its significance, it frequently appears as a benchmark problem in the literature[72,74]. In our experiments, we consider the generalized cluster Hamiltonian

$$H = \sum_{j=1}^{n} \left( Z_j - j_1 X_j X_{j+1} - j_2 X_{j-1} Z_j X_{j+1} \right), \tag{3}$$

where $n$ is the number of qubits, $X_i$ and $Z_i$ are Pauli operators acting on the $i^{\text{th}}$ qubit, and $j_1$ and $j_2$ are coupling strengths. Specifically, we

classify states according to which one of four symmetry-protected topological phases they display. As demonstrated in ref. 75, and depicted in Fig. 2, the ground-state phase diagram comprises the phases: (I) symmetry-protected topological, (II) ferromagnetic, (III) anti-ferromagnetic, and (IV) trivial.

The learning task we undertake involves identifying the correct quantum phase given the ground state of the generalized cluster Hamiltonian for some choice of $(j_1, j_2)$. We generate a training set $S = \{(|\psi_i\rangle, y_i)\}_{i=1}^N$ by sampling coupling coefficients uniformly at random in the domain $j_1, j_2 \in [-4, 4]$, with $N$ being the number of training data points, $|\psi_i\rangle$ representing the ground state vectors of $H$ corresponding to the sampled $(j_1, j_2)$, and $y_i$ denoting the corresponding phase label among the aforementioned phases. In particular, labels are length-two bit strings $y_i \in \{(0, 0), (0, 1), (1, 0), (1, 1)\}$.

We employ the QCNN architecture presented in ref. 69 to address the classification problem. By adapting classical convolutional neural networks to a quantum setting, QCNNs are particularly well-suited for tasks involving spatial and temporal patterns, which makes this architecture a natural choice for phase classification problems. A unique feature of the QCNN architecture is the interleaving of convolutional and pooling layers. Convolutional layers consist of translation-invariant parameterized unitaries applied to neighboring qubits, functioning as filters between feature maps across different layers of the QCNN. Following the convolutional layer, pooling layers are introduced to reduce the dimensionality of the quantum state while retaining the relevant features of the data. This is achieved by measuring a subset of qubits and applying translationally invariant parameterized single-qubit unitaries based on the corresponding measurement outcomes. Thus, each pooling layer consistently reduces the number of qubits by a constant factor, leading to quantum circuits with logarithmic depth relative to the initial system size. These circuits share a structural similarity to the multiscale entanglement renormalization ansatz[76]. Nevertheless, in instances where the input state to the QCNN exhibits, e.g., a high degree entanglement, the efficient classical simulation of the circuit becomes infeasible.

The operation of a QCNN can be interpreted as a quantum channel $\mathcal{C}_\vartheta$ specified by parameters $\vartheta$, mapping an input state $\rho_{\text{in}}$ into an output state $\rho_{\text{out}}$, represented as $\rho_{\text{out}} = \mathcal{C}_\vartheta[\rho_{\text{in}}]$. Subsequently, the expectation value of a task-oriented Hermitian operator is measured, utilizing the resulting $\rho_{\text{out}}$.

Our implementation follows that presented in ref. 54. The QCNN maps an input state vector $|\psi\rangle$, consisting of $n$ qubits, into a 2-qubit output state. For the labeling function given the output state, we use the probabilities of the outcome of each bit string when the state is measured in the computational basis $(p_{00}, p_{01}, p_{10}, p_{11})$. In particular, we predict the label $\hat{y}$ according to the measurement outcome with the lowest probability according to

$$|\psi\rangle \mapsto (p_b)_{b \in \{0,1\}^2} \mapsto \hat{y} := \arg\min_{b \in \{0,1\}^2} p_b . \tag{4}$$

For each experiment repetition, we generate data from the corresponding distribution $\mathcal{D}$. For training, we use the loss function

$$\ell(\vartheta; (|\psi\rangle, y)) := \langle y|(\mathcal{C}_\vartheta[|\psi_i\rangle\langle\psi_i|])|y\rangle . \tag{5}$$

This classification rule and loss function, which involve selecting the outcome with the lowest probability, was already utilized in ref. 54. The authors found that employing this seemingly counter-intuitive loss function lead to good generalization performance. Thus, given a training set $S \sim \mathcal{D}^N$, we minimize the empirical risk

$$\hat{R}_S(\vartheta) = \frac{1}{N} \sum_{i=1}^N \langle y_i|(\mathcal{C}_\vartheta[|\psi_i\rangle\langle\psi_i|])|y_i\rangle . \tag{6}$$

We consider three ways of altering the original data distribution $\mathcal{D}_0$ from where data is sampled, namely: (a) data wherein true labels are replaced by random labels $\mathcal{D}_1$, (b) randomization of only a fraction $r \in [0, 1]$ of the data, mixing real and corrupted labels in the same distribution $\mathcal{D}_r$, and (c) replacing the input quantum states with random states $\mathcal{D}_{\text{st}}$, instead of randomizing the labels. In each of these randomization experiments, the generalization gap and the risk functionals are defined according to the relevant distribution $\hat{\mathcal{D}} \in \{\mathcal{D}_1, \mathcal{D}_r, \mathcal{D}_{\text{st}}\}$. In all cases, the correlations between states and labels are gradually lost, which means we can control how much signal there is to be learned. In experiments where data-label correlations have vanished entirely, learning is impossible. One could expect the impossibility of learning to manifest itself during the training process, e.g., through lack of convergence. We observe that training the QCNN model on random data results in almost perfect classification performance on the training set. At face value, this means the QCNN is able to memorize noise.

In the following experiments, we approximate the expected risk $R$ with an empirical risk $\hat{R}_T$ using a large test set $T$. This test set is sampled independently from the same distribution as the training set $S$. In particular, the test set contains 1000 points for all the experiments, $T \sim \mathcal{D}^{1000}$.

Additionally, we report our results using the probability of error, which is further elucidated below. Consequently, we employ the term error instead of risk. Henceforth, we refer to test accuracy and test error as accurate proxies for the true accuracy and expected risk, respectively. All our experiments follow a three-step process:

1. Create a training set $S \sim \mathcal{D}^N$ and a test set $T \sim \mathcal{D}^{1000}$.
2. Find a function $f$ that approximately minimizes the empirical risk of Eq. (6).
3. Compute the training error $\hat{R}_S(f)$, test error $\hat{R}_T(f)$, and the empirical generalization gap $\text{gen}_T(f) = |\hat{R}_T(f) - \hat{R}_S(f)|$.

For ease of notation, we shall employ $\text{gen}(f)$ instead of $\text{gen}_T(f)$ while discussing the generalization gap without reiterating its empirical nature.

**Random labels.** We start our randomization tests by drawing data from $\mathcal{D}_1$, wherein the true labels have been replaced by random labels sampled uniformly from $\{(0, 0), (0, 1), (1, 0), (1, 1)\}$. In order to sample from $\mathcal{D}_1$, a labeled pair can be obtained from the original data distribution $(|\psi\rangle, y) \sim \mathcal{D}_0$, after which the label $y$ can be randomly replaced. In this experiment, we have employed QCNNs with varying numbers of qubits $n \in \{8, 16, 32\}$. For each qubit number, we have generated training sets with different sizes $N \in \{5, 8, 10, 14, 20\}$ for both random

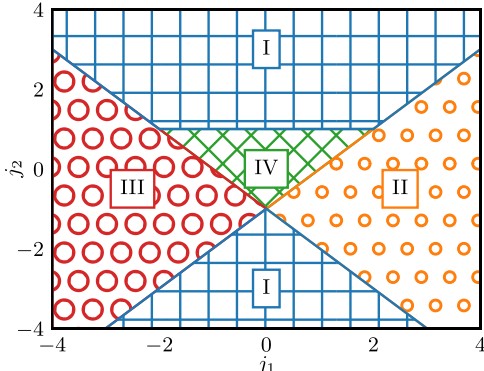

**Fig. 2 | Phase diagram of the generalized cluster Hamiltonian.** The ground-state phase diagram of the Hamiltonian of Eq. (3). It comprises the phases: (I) symmetry-protected topological, (II) ferromagnetic, (III) anti-ferromagnetic, and (IV) trivial.

and real labels. The models were trained individually for each $(n, N)$ combination.

In Fig. 3a, we illustrate the results obtained when fitting random and real labels, as well as random states (discussed later). Each data point in the figure represents the average generalization gap achieved for a fixed training set size $N$ for the different qubit numbers $n$. We observe a large gap for the random labels, close to 0.75, which should be seen as effectively maximal: perfect training accuracy and the same test accuracy as random guessing would yield. This finding suggests that the QCNN can be adjusted to fit the random labels in the training set, despite the labels bearing no correlation to the input states. As the training set sizes increase, since the capacity of the QCNN is fixed, achieving a perfect classification accuracy for the entire training set becomes increasingly challenging. Consequently, the generalization gap diminishes. It is worth noting that a decrease in training accuracy is also observed for the true labeling of data[54].

**Corrupted labels.** Next to the randomization of labels, we further investigate the QCNN fitting behavior when data come with varying levels of label corruption $\mathcal{D}_r$, ranging from no labels being altered ($r = 0$) to all of them being corrupted ($r = 1$). The experiments consider different number of training points $N \in \{4, 6, 8\}$, and a fixed number of qubits $n = 8$. For each combination of $(n, N)$, we start the experiments with no randomized labels ($r = 0$). Then, we gradually increase the ratio of randomized labels until all labels are altered, that is, $r \in \{0, 1/N, 2/N, ..., 1\}$. Figure 3b shows the test error after convergence. In all repetitions, this experiment reaches 100% training accuracy. We observe a steady increase in the test error as the noise level intensifies. This suggests that QCNNs are capable of extracting the remaining signal in the data while simultaneously fitting the noise by brute force. As the label corruption approaches 1, the test error converges to 75%, corresponding to the performance of random guessing.

The inset in Fig. 3b focuses on the experiments conducted with $N = 6$ training points. In particular, we examine the relationship between the learning speed and the ratio of random labels. The plot shows an average over five experiment repetitions. Remarkably, each individual run exhibits a consistent pattern: the training error initially remains high, but it converges quickly once the decrease starts. This behavior was also reported for classical neural networks[62]. The precise moment at which the training error begins to decrease seems to be heavily dependent on the random initialization of the parameters. However, it also relates to the signal-to-noise ratio $r$ in the training data. Notably, we observe a long and stable plateau for the intermediate cases $r = 1/3$ and $r = 2/3$, roughly halfway between the starting training error and zero. This plateau represents an average between those runs where the rapid decrease has not yet started and those where the convergence has already been achieved, leading to significant variance. Interestingly, in the complete absence of correlation between states and labels ($r = 1$), the QCNN, on average, perfectly fits the training data even slightly faster than for the real labels ($r = 0$).

**Random states.** In this scenario, we introduce randomness to the input ground state vectors rather than to the labels. Our goal is to introduce a certain degree of randomization into the quantum states while preserving some inherent structure in the problem. To achieve this, we define the data distribution $\mathcal{D}_{st}$ for the random quantum states in a specific manner instead of just drawing pure random states uniformly.

To sample data from $\mathcal{D}_{st}$, we first draw a pair from the original distribution $(|\psi\rangle, y) \sim \mathcal{D}_0$, and then we apply the following transformation to the state vector $|\psi\rangle$: We compute the mean $\mu_\psi$ and variance

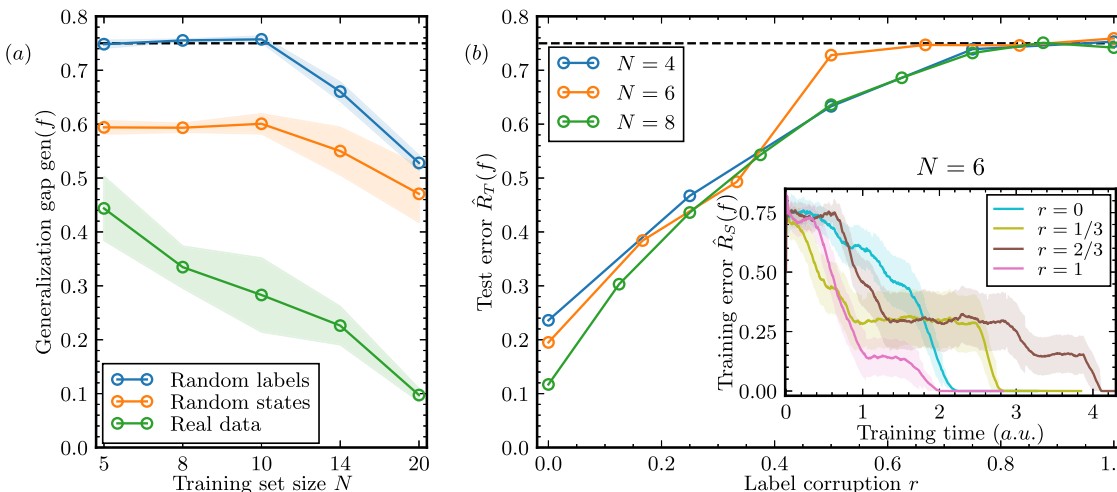

**Fig. 3 | Randomization tests for quantum phase recognition. a** Generalization gap as a function of the training set size achieved by the quantum convolutional neural network (QCNN) architecture. The QCNN is trained on real data, random label data, and random state data. The horizontal dashed line is the largest generalization gap attainable, characterized by zero training error and test error equal to random guessing (0.75 due to the task having four possible classes). The shaded area corresponds to the standard deviation across different experiment repetitions. For the real data and random labels, we employed 8, 16, and 32 qubits, while for the random states, we employed 8, 10, and 12 qubits. We observe that both random labels and random states exhibit a similar trend in the generalization gap, with a slight discrepancy in height due to the different relative frequencies of the four classes under the respective randomization protocols. In both cases, the test accuracy fails to surpass that of random guessing. Notably, the largest generalization gap occurs in the random labels experiments when using a training set of up to size $N = 10$, highlighting the memorization capacity of this particular QCNN. The training with uncorrupted data yields behavior in accordance with previous results[54]. **b** Test error as a function of the ratio of label corruption after training the QCNN on training sets of size $N \in 4, 6, 8$ and $n = 8$. The plot illustrates the interpolation between uncorrupted data ($r = 0$) and random labels ($r = 1$). As the label corruption approaches 1, the test accuracy drops to levels of random guessing. The dependence between the test error and label corruption reveals the ability of the QCNN to extract remaining signal despite the noise in the initial training set. The inset focuses on the case $N = 6$. It conveys the optimization speed for four different levels of corruption, namely, 0, 2, 4, and 6 out of 6 labels being corrupted, and provides insights into the average convergence time. The shaded area denotes the variance over five experiment repetitions with independently initialized QCNN parameters. Surprisingly, on average, fitting completely random noise takes less time than fitting unperturbed data. This phenomenon emphasizes that QCNNs can accurately memorize random data.

$\sigma_\psi$ of its amplitudes and then sample new amplitudes randomly from a Gaussian distribution $\mathcal{N}(\mu_\psi, \sigma_\psi)$. After the new amplitudes are obtained, we normalize them. The random state experiments were performed with varying numbers of qubits $n \in \{8, 10, 12\}$ and training set sizes $N \in \{5, 8, 10, 14, 20\}$.

In Fig. 3a, we show the results for fitting random input states, together with the random and real label experiment outcomes. The empirical generalization gaps achieved by the QCNN for random states exhibit a similar shape to those obtained for random labels. Indeed, a slight difference in the relative occurrences of each of the four classes leads to improved performance by biased random guessing. We observe that the QCNN can perfectly fit the training set for few data, and then the generalization gap decreases, analogously to the scenario with random labels.

The case of random states presents an intriguing aspect. The QCNN architecture was initially designed to unveil and exploit local correlations in input quantum states[69]. However, our randomization protocol in this experiment removes precisely all local information, leaving only global information from the original data, such as the mean and the variance of the amplitudes. This was not the case in the random labels experiment, where the input ground states remained unaltered while only the labels were modified. The ability of the QCNN to memorize random data seems to be unaffected despite its structure to exploit local information.

## Implications

Our findings indicate that novel approaches are required in studying the capabilities of quantum neural networks. Here, we elucidate how our experimental results fit the statistical learning theoretic framework. The main goal of machine learning is to find the expected risk minimizer $f^{\mathrm{opt}}$ associated with a given learning task,

$$f^{\mathrm{opt}} := \arg\min_{f \in \mathcal{F}} R(f). \qquad (7)$$

However, given the unknown nature of the complete data distribution $\mathcal{D}$, the evaluation of $R$ becomes infeasible. Consequently, we must resort to its unbiased estimator, the empirical risk $\hat{R}_S$. We let an optimization algorithm obtain $f$, an approximate empirical risk minimizer

$$f^* \approx \arg\min_{f \in \mathcal{F}} \hat{R}_S(f). \qquad (8)$$

Nonetheless, although $\hat{R}_S(f)$ is an unbiased estimator for $R(f)$, it remains uncertain whether the empirical risk minimizer $f$ will yield a low expected risk $R(f)$. The generalization gap gen($f$) then comes in as the critical quantity of interest, quantifying the difference in performance on the training set $\hat{R}_S(f)$ and the expected performance on the entire domain $R(f)$.

In the literature, extensive efforts have been invested in providing robust guarantees on the magnitude of the generalization gap of QML models through so-called generalization bounds[45–52,54,58,59,65]. These theorems assert that under reasonable assumptions, the generalization gap of a given model can be upper bounded by a quantity that can depend on various parameters. These include properties of the function family, the optimization algorithm used, or the data distribution. The derivation of a generalization bound for a learning model typically involves rigorous mathematical calculations and often considers restricted scenarios. Many results in the literature fit the following template:

**Generic uniform generalization bound.** Let $\mathcal{F}$ be a hypothesis class, and let $\mathcal{D}$ be any data-generating distribution. Let $R$ be a risk functional associated to $\mathcal{D}$, and $\hat{R}_S$ its empirical version, for a given set of $N$ labeled data: $S \sim \mathcal{D}^N$. Let $C(\mathcal{F})$ be a complexity measure of $\mathcal{F}$. Then, for any

function $f \in \mathcal{F}$, the generalization gap gen($f$) can be upper bounded, with high probability, by

$$\mathrm{gen}(f) \leq g_{\mathrm{unif}}(\mathcal{F}), \qquad (9)$$

where usually $g_{\mathrm{unif}}(\mathcal{F}) \in \mathcal{O}(\mathrm{poly}(C(\mathcal{F}), 1/N))$ is given explicitly. We make the dependence of $g_{\mathrm{unif}}$ on $N$ implicit for clarity. The high probability is taken with respect to repeated sampling from $\mathcal{D}$ of sets $S$ of size $N$.

We refer to these as uniform generalization bounds by virtue of them being equal for all elements $f$ in the class $\mathcal{F}$. Also, these bounds apply irrespective of the probability distribution $\mathcal{D}$. There exists a singular example that does not fit the template in ref. 57. In this particular case, the authors introduce a robustness-based complexity measure, resulting in a bound that depends on both the data distribution and the learned hypothesis, albeit very indirectly. As a result, it presents difficulties for quantitative predictions.

The usefulness of uniform generalization bounds lies in their ability to provide performance guarantees for a model before undertaking any computationally expensive training. Thus, it becomes of interest to identify ranges of values for $C(\mathcal{F})$ and $N$ that result in a diminishing or entirely vanishing generalization gap (such as the limit $N \to \infty$). These bounds usually deal with asymptotic regimes. Thus it is sometimes unclear how tight their statements are for practical scenarios.

In cases where the risk functional is itself bounded, we can further refine the bound. For example, if we take $R^e$ to be the probability of error

$$R^e(f) = \mathbb{P}_{(x,y) \sim \mathcal{D}}[f(x) \neq y] \in [0,1], \qquad (10)$$

we can immediately say that, for any $f$, there is a trivial upper bound on the generalization gap gen($f$) $\leq 1$. Thus, the generalization bound could be rewritten as

$$\mathrm{gen}(f) \leq \min\{1, g_{\mathrm{unif}}(\mathcal{F})\}. \qquad (11)$$

This additional threshold renders the actual value of $g_{\mathrm{unif}}(\mathcal{F})$ of considerable significance.

We now have the necessary tools to discuss the results of our experiments properly. Randomizing the data simply involves changing the data-generating distribution, e.g., from the original $\mathcal{D}_0$ to a randomized $\hat{\mathcal{D}} \in \{\mathcal{D}_1, \mathcal{D}_r, \mathcal{D}_{\mathrm{st}}\}$. As we have just remarked, the r.h.s. of Eq. (9) does not change for different distributions, implying that the same upper bound on the generalization gap applies to both data coming from $\mathcal{D}_0$, or corrupted data from $\hat{\mathcal{D}}$. If data from $\hat{\mathcal{D}}$ is such that inputs and labels are uncorrelated, then any hypothesis cannot be better than random guessing in expectation. This results in the expected risk value being close to its maximum. For instance, in the case of the probability of error and a classification task with $M$ classes, if each input is assigned a class uniformly at random, then it must hold for any hypothesis $f$,

$$R^e(f) \approx 1 - \frac{1}{M}, \qquad (12)$$

indicating that the expected risk must always be large.

A large risk for a particular example does not generally imply a large generalization gap gen($f$) $\approx R^e(f)$. For instance, if a learning model is unable to fit a corrupted training set $S$, $\hat{R}_S^e(f) \approx R^e(f)$, then one would have a small generalization gap gen($f$) $\approx 0$. Conversely, for the generalization gap of $f$ to be large gen($f$) $\approx 1 - 1/M$, the learning algorithm must find a function that can actually fit $S$, with $\hat{R}_S^e(f) \approx 0$. Yet, even in this last scenario, the uniform generalization bound still applies.

Let us denote $N'$ the size of the largest training set $S$ for which we found a function $f_r$ able to fit the random data $\hat{R}_S^e(f_r) \approx 0$ (which leads to

a large generalization gap gen($f_r$) ≈ 1 − 1/$M$. Since the uniform generalization bound applies to all functions in the class $f \in \mathcal{F}$, we have found

$$g_{\text{unif}}(\mathcal{F}) \gtrsim 1 - \frac{1}{M} \quad (13)$$

as an empirical lower bound to the generalization bound. This reveals that the generalization bound is vacuous for training sets of size up to $N'$. Noteworthy is also that, further than $N'$, there is a regime where the generalization bound remains impractically large.

The strength of our results resides in the fact that we did not need to specify a complexity measure $C(\mathcal{F})$. Our empirical findings apply to every uniform generalization bound, irrespective of its derivation. This gives strong evidence for the need for a perspective shift to the study of generalization in quantum machine learning.

## Analytical results

In the previous section, we provided evidence that QNNs can accurately fit random labels in near-term experimental set-ups. Our empirical findings are restricted to the number of qubits and training samples we tested. While these limitations seem restrictive, they are actually the relevant regimes of interest, considering the empirical evidence. In this section, we formally study the memorization capability of QML models of arbitrary size, beyond the NISQ era, in terms of finite sample expressivity. Our goal is to establish sufficient conditions for demonstrating how QML models could fit arbitrary training sets, and not to establish that it is always possible in a worst-case scenario.

Finite sample expressivity refers to the ability of a function family to memorize arbitrary data. In general, expressivity is the ability of a hypothesis class to approximate functions in the entire domain $\mathcal{X}$. Conversely, finite sample expressivity studies the ability to approximate functions on fixed-size subsets of $\mathcal{X}$. Although finite sample expressivity is a weaker notion of expressivity, it can be seen as a stronger alternative to the pseudo-dimension of a hypothesis family[45,65].

The importance of finite sample expressivity lies in the fact that machine learning tasks always deal with finite training sets. Suppose a given model is found to be able to realize any possible labeling of an available training set. Then, reasonably one would not expect the model to learn meaningful insights from the training data. It is plausible that some form of learning may still occur, albeit without a clear understanding of the underlying mechanisms. However, under such circumstances, uniform generalization bounds would inevitably become trivial.

**Theorem 1.** (Finite sample expressivity of quantum circuits). Let $\rho_1, \ldots, \rho_N$ be unknown quantum states on $n \in \mathbb{N}$ qubits, with $N \in \mathcal{O}(\text{poly}(n))$, and let $W$ be the Gram matrix

$$[W]_{i,j} = \text{tr}(\rho_i \rho_j). \quad (14)$$

If $W$ is well-conditioned, then, for any $y_1, \ldots, y_N \in \mathbb{R}$ real numbers, we can construct a quantum circuit of depth poly($n$) as an observable $\mathcal{M}_y$ such that

$$\text{tr}(\rho_i \mathcal{M}_y) = y_i. \quad (15)$$

The proof is given in Supplementary Note 1. Theorem 1 gives us a constructive approach to, given a finite set of quantum states and real labels, find a quantum circuit that produces each of the labels as the expectation value for each of the input states. This should give an intuition for why QML models seem capable of learning random labels and random quantum states. Nevertheless, as stated, the theorem falls

short in applying specifically to PQCs. The construction we propose requires query access to the set of input states every time the circuit is executed. We estimate the values $\text{tr}(\rho_i \rho_j)$ employing the SWAP test. The circuit that realizes the SWAP test bears little relation to usual QML ansätze. Ideally, if possible, one should impose a familiar PQC structure and drop the need to use the input states.

Next, we propose an alternative, more restricted version of the same statement, keeping QML in mind as the desired application. For it, we need a sense of distinguishability of quantum states.

**Definition 1.** (Distinguishability condition). We say $n$-qubit quantum states $\rho_1, \ldots, \rho_N$ fulfill the distinguishability condition if we can find intermediate states $\rho_i \mapsto \hat{\rho}_i$ based on some generic quantum state approximation protocol such that they fulfill the following:
1. For each $i \in [N]$, $\hat{\rho}_i$ is efficiently preparable with a PQC.
2. The matrix $\hat{W}$ can be efficiently constructed, with entries

$$\hat{W}_{i,j} = \text{tr}(\rho_i \hat{\rho}_j). \quad (16)$$

3. The matrix $\hat{W}$ is well-conditioned.

Notable examples of approximation protocols are those inspired by classical shadows[77] or tensor networks[78]. For instance, similarly to classical shadows, one could draw unitaries from an approximate poly($n$)-design using a brickwork ansatz with poly($n$)-many layers of i.i.d. Haar random 2-local gates. For a given quantum state $\rho$, one produces several pairs $(U, b)$ where $U$ is the randomly drawn unitary and $b$ is the bit-string outcome after performing a computational basis measurement of $U\rho U^\dagger$, and one refers to each individual pair as a snapshot. Notice that this approach does not follow exactly the traditional classical shadows protocol. Our end goal is to prepare the approximation as a PQC, rather than utilizing it for classical simulation purposes. In particular, we do not employ the inverse measurement channel, since that would break complete positivity and thus the corresponding approximation would not be a quantum state. For each snapshot, one can efficiently prepare the corresponding quantum state $U^\dagger |b\rangle\langle b| U$ by undoing the unitary that was drawn after preparing the corresponding computational basis state vector $|b\rangle$. Given a collection of snapshots $\{(U_1, b_1), \ldots, (U_M, b_M)\}$, an approximation protocol would consist of preparing the mixed state $\frac{1}{M}\sum_{m=1}^{M} U_m^\dagger |b_m\rangle\langle b_m| U_m$. Since each $b_m$ is prepared with at most $n$ Pauli-$X$ gates and each $U_m$ is a brickwork PQC architecture, this approximation protocol fulfills the restriction of efficient preparation from Definition 1. Whether or not this or any other generic approximation protocol is accurate enough for a specific choice of quantum states we discuss in Methods' sub-section "Analytical methods". There, we present an algorithm in Box 1 together with its correctness statement as Theorem 3. Given the input states $\rho_1, \ldots, \rho_N$ Box 1 moreover allows to combine several quantum state approximation protocols in order to produce a well-conditioned matrix of inner products $\hat{W}$.

**Theorem 2.** (Finite sample expressivity of PQCs) Let $\rho_1, \ldots, \rho_N$ be unknown quantum states on $n \in \mathbb{N}$ qubits, with $N \in \mathcal{O}(\text{poly}(n))$, and fulfilling the distinguishability condition of Definition 1. Then, we can construct a PQC of poly($n$) depth as a parameterized observable $\hat{\mathcal{M}}(\vartheta)$ such that, for any $y = (y_1, \ldots, y_N) \in \mathbb{R}$ real numbers, we can efficiently find a specification of the parameters $\vartheta_y$ such that

$$\text{tr}(\rho_i \hat{\mathcal{M}}(\vartheta_y)) = y_i. \quad (17)$$

The proof is given in Supplementary Note 2, which uses ideas reminiscent to the formalism of linear combinations of unitary operations[79]. With Theorem 2, we understand that PQCs can produce any labeling of arbitrary sets of quantum states, provided they fulfill our distinguishability condition.

Notice that Definition 1 is needed for the correctness of Theorem 2. We require knowledge of an efficient classical description of the quantum states for two main reasons. On the one hand, PQCs are the object of our study. Hence, we need to prepare the approximation efficiently as a PQC. In addition, on the other hand, the distinguishability condition is also enough to prevent us from running into computation-complexity bottle-necks, like those arising from the distributed inner product estimation results in ref. 80.

## Discussion

We next discuss the implications of our results and suggest research avenues to explore in the future. We have shown that quantum neural networks (QNNs) can fit random data, including randomized labels or quantum states. We provided a detailed explanation of how to place our findings in a statistical learning theory context. We do not claim that uniform generalization bounds are wrong or that any prior results are false. Instead, we show that the statements of theorems that fit our generic uniform template must be vacuous for the regimes where the models are able to fit a large fraction of random data. We have brought the randomization tests of ref. 62 to the quantum level. We have selected one of the most promising QML architectures for our experiments, known as the quantum convolutional neural network (QCNN). We have considered the task of classifying quantum phases of matter, which is a state-of-the-art application of QML.

Our numerical results suggest that we must reach further than uniform generalization bounds to fully understand quantum machine learning (QML) models. In particular, experiments like ours immediately problematize approaches based on complexity measures like the VC dimension, the Rademacher complexity, and all their uniform relatives. To the best of our knowledge, essentially all generalization bounds derived for QML so far are of the uniform kind. Therefore, our findings highlight the need for a perspective shift in generalization for QML. In the future, it will be interesting to conduct causation experiments on QNNs using non-uniform generalization measures. Promising candidates for good generalization measures in QML include the time to convergence of the training procedure, the geometric sharpness of the minimum the algorithm converged to, and the robustness against noise in the data[81].

The structure of the QCNN, with its equivariant and pooling layers, results in an ansatz with restricted expressivity. Its core features, including intermediate measurements, parameter-sharing, and logarithmic depth, make the QCNN a smaller model than other, deeper PQCs, like a brickwork ansatz with completely unrestricted parameters. In the language of traditional statistical learning, this translates to higher bias and lower variance. Consequently, for the same task, the QCNN tends towards underfitting, posing a greater challenge in achieving perfect fitting of the training set compared to a more expressive model. As a result, the QCNN is anticipated to exhibit better generalization behavior when compared to the usual hardware-efficient ansätze[82]. The QCNN thus is assigned lower generalization bounds than other larger models, due to the higher variance of the latter. Therefore, our demonstration that uniform generalization bounds applied to the QCNN family are trivially loose immediately implies that the same bounds applied to less restricted models must also be vacuous. Stated differently, our findings for a small model, the QCNN, inherently apply to all larger models, including the hardware-efficient ansatz. Furthermore, our study adds to the evidence supporting the need for a proper understanding of symmetries and equivariance in QML[58,83–85].

In addition to our numerical experiments, we have analytically shown that polynomially-sized QNNs are able to fit arbitrary labeling of data sets. This seems to contradict claims that few training data are provably sufficient to guarantee good generalization in QML, raised e.g. in ref. 54. Our analytical and numerical results do not preclude the possibility of good generalization with few training data but rather

indicate we cannot guarantee it with arguments based on uniform generalization bounds. The reasons why successful generalization might occur have yet to be discovered.

We employ the rest of this section to comment on the significance of our randomization experiments, and also describe the parallelisms and differences between our work and the seminal ref. 62, which served as the basis for our experimental design. In particular, two primary factors warrant consideration: the size of the model, and the size of the training set. In our learning task, quantum phase recognition problem for systems of up to 32 qubits, we use training sets comprised of up to 20 labeled pairs. In the following paragraphs we elucidate whether these should be considered large or small; capable of overfitting or memorizing; and whether the results of our experiments are due to finite sample size artifacts.

Upon first glance, the training set sizes employed in our randomization experiments may seem relatively small. However, it is essential to consider the randomization study within its relevant context. As previously mentioned, good generalization performance has been reported in QML, particularly for classifying quantum phases of matter using a QCNN architecture[54]. At present, this combination of model and task is also among the best leading approaches concerning generalization within the QML literature. The key fact is that our randomization tests use the same training set sizes as the original experiments which reported good generalization performance. The question whether the randomization results are caused by the relative ease to find patterns that fit the given labels from the small set of data is ruled out by the fact that these small set sizes suffice to solve the original problem. If the QCNN were able to fit the random data only because of finite sample size artifacts, we would anticipate the expected risk and the generalization gap to be considerably large even for the original data. Given our observation of successful generalization for data sampled from the original distribution, we conclude that these training sets are not too small, but rather large enough.

Both our study and ref. 62 have in common that the learning models considered were regarded as among the best in terms of generalization for state-of-the-art benchmark tasks. Also, the randomization experiments in both cases employed datasets taken from state-of-the-art experiments of the time. Yet, and in spite of the similarities, it is imperative to recognize that the learning models employed in these studies are fundamentally different. They not only operate on distinct computing platforms of a physically different nature, but also the functions produced by neural networks are typically different from those produced by parameterized quantum circuits. As a consequence, caution is warranted in expecting these two different learning models to behave equally when faced with randomization experiments based on unrelated learning tasks. The fact that the quantum and classical learning models display similar results should not be taken for granted.

A key distinction lies in the notion of overparameterization, which plays a critical role in classical machine learning. It is important to distinguish the notion of overparameterization in classical ML from the recently introduced definition of overparameterization in QML[42], which under the same name, deals with different concepts. The deep networks studied in ref. 62 have far more parameters than both the dimension of the input image and the training set size. This brings us to refer to these as large models. Conversely, we argue that the QCNN qualifies as a small model. Although the number of parameters in the considered architectures is larger than the size of the training sets, they exhibit a logarithmic scaling with the number of qubits. Meanwhile, the number of dimensions of the quantum states scales exponentially. Hence, it is inappropriate to categorize the models we have investigated as large in the same way as the classical models in ref. 62. We find the ability of small quantum learning models to fit random data as unexpected, as witnessed by the many works on uniform

generalization bounds for quantum models published during the aftermath of ref. 62. This observation reveals a promising research direction: not only must we rethink our approach to studying generalization in QML, but we must also recognize that the mechanisms leading to successful generalization in QML may differ entirely from those in classical machine learning. On a higher level, this work exemplifies the necessity of establishing connections between the literature on classical machine learning and the evolving field of quantum machine learning.

## Methods
### Numerical methods
This section provides a comprehensive description of our numerical experiments, including the computation techniques employed for the random and real label implementations, as well as the random state and partially corrupted label implementations.

**Random and real label implementations.** The test and training ground state vectors $|\psi_i\rangle$ of the cluster Hamiltonian in Eq. (3) have been obtained as variational principles over matrix product states in a reading of the density matrix renormalization group ansatz[86] through the software package `Quimb`[87]. We have utilized the matrix product state backend from `TensorCircuit`[88] to simulate the quantum circuits. In particular, a bond dimension of $\chi = 40$ was employed for the simulations of 16- and 32-qubit QCNNs. We find that further increasing the bond dimension does not lead to any noticeable changes in our results.

**Random state and partially-corrupted label implementations.** In this scenario, the test and training ground state vectors $|\psi_i\rangle$ were obtained directly diagonalizing the Hamiltonian. Note that our QCNN comprised a smaller number of qubits for these examples, namely, $n \in \{8, 10, 12\}$. The simulation of quantum circuits was performed using `Qibo`[89], a software framework that allows faster simulation of quantum circuits.

For all implementations, the training parameters were initialized randomly. The optimization method employed to update the parameters of the QCNN during training is the `CMA-ES`[90], a stochastic, derivative-free optimization strategy. The code generated under the current study is also available in ref. 91.

### Analytical methods
Here, we shed light on the practicalities of Definition 1, a requirement for our central Theorem 2. The algorithm in Box 1 allows for several approximation protocols to be combined to increase the chances of fulfilling the assumptions of Definition 1. Indeed, we can allow for the auxiliary states $\hat{\rho}_1, \ldots, \hat{\rho}_N$ to be linear combinations of several approximation states while staying in the mindset of Definition 1. Then, we can cast the problem of finding an optimal weighting for the linear combination as a linear optimization problem with a positive semi-definite constraint.

With Theorem 3, we can assess the distinguishability condition of Definition 1 for specific states $\rho_1, \ldots, \rho_N$ and specific approximation protocols. Theorem 3 also considers the case where different approximation protocols are combined, which does not contradict the requirements of Theorem 2.

**Theorem 3.** (Conditioning as a convex program 1). Let $\rho_1, \ldots, \rho_N$ be unknown, linearly-independent quantum states on $n$ qubits, with $N \in \mathcal{O}(\text{poly}(n))$. For any $i \in [N]$, let $\sigma^i = (\sigma_1^i, \ldots, \sigma_m^i)$ be approximations of $\rho_i$, each of which can be efficiently prepared using a PQC. Assume the computation of $\text{tr}(\rho_i \sigma_k^j)$ in polynomial time for any choice of $i, j$ and $k$. Call $\sigma = (\sigma^1, \ldots, \sigma^N)$. The real numbers $\alpha = (\alpha_{i,k})_{i \in [N], k \in [m]} \in \mathbb{R}^{Nm}$ define

---

**BOX 1**
# Convex optimization state approximation

**Require:**
1: $\rho = (\rho_1, \ldots, \rho_N)$ ▷ Quantum states
2: $A = (A_1, \ldots, A_m)$ ▷ State approximation algorithms
3: $\kappa$ ▷ Condition number

**Ensure:** $\alpha$ such that $\hat{\mathbf{W}}$ is well-conditioned if possible, 0 otherwise.

4:
5: **for** $i \in [N], k \in [m]$ **do**
6: $\quad \sigma_k^i \leftarrow A_k(\rho_i)$
7: **end for**
8:
9: $\sigma \leftarrow (\sigma_k^i)_{i \in [N], k \in [m]}$
10:
11: $\alpha \leftarrow \text{SDP}(\rho, \sigma, \kappa)$ ▷ From proof of Theorem 3
12:
13: **if** SDP fails **then**
14: $\quad$ **return** 0 ▷ No suitable $\alpha$ found
15:
16: **else**
17: $\quad$ **return** $\alpha$ ▷ $\hat{\mathbf{W}}$ well-conditioned
18: **end if**

---

the auxiliary states $\hat{\rho}_1, \ldots, \hat{\rho}_N$ as

$$\hat{\rho}_i(\alpha; \sigma^i) = \sum_{k=1}^m \alpha_{i,k} \sigma_i^k, \quad (18)$$

and the matrix of inner products $\hat{W}(\alpha; \sigma)$ with entries

$$\left[\hat{W}(\alpha; \sigma)_{i,j}\right]_{i,j \in [N]} := \text{tr}\left(\rho_i \hat{\rho}_j(\alpha; \sigma^j)\right) \quad (19)$$

$$= \sum_{k=1}^m \alpha_{j,k} \text{tr}\left(\rho_i \sigma_k^j\right). \quad (20)$$

Then, $\|\hat{W}(\alpha; \sigma)\| \leq N$. Further, one can then decide in polynomial time whether, given $\rho_1, \ldots, \rho_N$, $\sigma$, and $\kappa \in \mathbb{R}$, there exists a specification of $\alpha \in \mathbb{R}^{Nm}$ such that $\hat{W}(\alpha; \sigma)$ is well-conditioned in the sense that $\|\hat{W}(\alpha; \sigma)^{-1}\|^{-1} \geq \kappa$. And, if there exists such a specification, a convex semi-definite problem (SDP) outputs an instance of $\alpha \leftarrow \text{SDP}(\rho, \sigma, \kappa)$ for which $\hat{W}$ is well-conditioned. If it exists, one can also find in polynomial time the $\alpha$ with the smallest $\|\cdot\|_{l_1}$ or $\|\cdot\|_{l_2}$ norm.

**Proof.** The inequality $\|\hat{W}(\alpha; \sigma)\| \leq N$ follows from Gershgorin's circle theorem[92], given that all entries of $\hat{W}$ are bounded between $[0, 1]$. In particular, the largest singular value of the matrix $\hat{W}$ reaches the value $N$ when all entries are 1.

The expression

$$\hat{W}_{i,j} = \sum_{k=1}^m \alpha_{j,k} \text{tr}\left(\rho_i \sigma_k^j\right). \quad (21)$$

is a linear constraint on $\alpha$ and $\hat{W}$, for $i, j \in [N]$, while

$$\kappa \mathbb{I} \leq \hat{W} \leq N \mathbb{I} \quad (22)$$

in matrix ordering is a positive semi-definite constraint. $\hat{W} \leq N\mathbb{I}$ is equivalent with $\| \hat{W} \| \leq N$, while $\kappa\mathbb{I} \leq \hat{W}$ means that the smallest singular value of $\hat{W}$ is lower bounded by $\kappa$, being equivalent with

$$\| \hat{W}(\alpha;\sigma)^{-1} \|^{-1} \leq \kappa , \tag{23}$$

for an invertible $\hat{W}(\alpha;\sigma)$. The test whether such a $\hat{W}$ is well-conditioned hence takes the form of a semi-definite feasibility problem[93]. One can additionally minimize the objective functions

$$\alpha \mapsto \| \alpha \|_{l_1} \tag{24}$$

and

$$\alpha \mapsto \| \alpha \|_{l_2} , \tag{25}$$

both again as linear or convex quadratic and hence semi-definite problems. Overall, the problem can be solved as a semi-definite problem, that can be solved in a run-time with low-order polynomial effort with interior point methods. Duality theory readily provides a rigorous certificate for the solution[93].

In the proof, we refrain from explicitly specifying the definition of $\sigma$ in relation to the original states $\rho_1, ..., \rho_N$. Indeed, the success criterion is that the resulting matrix $\hat{W}$ is well-conditioned. As a sufficient condition, we could have required both that the Gram matrix $W_{i,j} = \mathrm{tr}(\rho_i\rho_j)$ is well-conditioned, and that for each $i \in [N]$ there is at least one $k \in [m]$ such that $\sigma_i^k$ is close to $\rho_i$ for some distance metric. Under this condition, we would expect $\hat{W}$ to be well-conditioned. Nonetheless, this condition is not necessary. In general, it is plausible that each of the states $\hat{\rho}_i$ constructed from $\sigma$ are not close to each of the original states $\rho_i$, resulting in $\hat{W}$ not being close to $W$, while $\hat{W}$ still being well-conditioned. In this situation, the construction from Theorem 2 still holds.

We propose using Box 1 to construct the optimal auxiliary states $\hat{\rho}_1, ... \hat{\rho}_N$, given the unknown input states $\rho_1, ..., \rho_N$ and a collection of available approximation protocols $A_1, ..., A_m$. The algorithm produces an output of either 0 in cases where no combination of the approximation states satisfies the distinguishability condition, or it provides the weights $\alpha$ necessary to construct the auxiliary states as a sum of approximation states. In Theorem 3, we prove the correctness of the algorithm.

The construction of $\sigma$ from the input states $\rho_1, ..., \rho_N$ plays an intuitive role in the success of Box 1. Let us consider two scenarios. First, we assume the Gram matrix of the initial states $W$ is well-conditioned, and that for each $i \in [N]$ there is at least one $k \in [m]$ such that $\rho_i = \sigma_i^k$. In this instance, there exists at least one specification of real values $\alpha$ for which $\hat{W}$ is well-conditioned. It suffices to set $\alpha_{j,k} = \delta_{j,k}$, the latter denoting the Kronecker delta. This guarantees that the algorithm in Box 1 outputs a satisfactory $\alpha$ (potentially of minimal norm) in polynomial time. Conversely, we now consider a scenario where the approximation protocols employed to construct $\sigma$ all yield failures, resulting in $\sigma_i^k = |0\rangle\langle 0|$ for all $i \in [N]$ and $k \in [m]$. In this case, there is no choice of $\alpha$ for which $\hat{W}$ is well conditioned and Box 1 necessarily outputs 0, also within polynomial time.

We refer to the proof of Theorem 2, in Supplementary Note 2, for an explanation of how to construct the intermediate states $\hat{\rho}_i$ as a linear combination of auxiliary states $\sigma^i$ without giving up the PQC framework.

## Data availability
The data used in this study are available in the Zenodo database in ref. 91.

## Code availability
The code used in this study is available in the Zenodo database in ref. 91.

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

## Acknowledgements
We would like to thank Matthias C. Caro, Vedran Dunjko, Johannes Jakob Meyer, and Ryan Sweke for useful comments on an earlier version of this manuscript and Christian Bertoni, José Carrasco, and Sofiene Jerbi for insightful discussions. We also acknowledge the BMBF (MUNIQC-Atoms, Hybrid), the BMWK (EniQmA, PlanQK), the QuantERA (HQCC), the Quantum Flagship (PasQuans2), the MATH+ Cluster of Excellence, the DFG (CRC 183, B01), the Einstein Foundation (Einstein Research Unit on Quantum Devices), and the ERC (DebuQC) for financial support.

## Author contributions
The project has been conceived by C.B.-P. Experimental design has been laid out by E.G.-F. Analytical results have been proven by E.G.-F. and J.E. Numerical experiments have been performed by C.B.-P. The project has been supervised by C.B.-P. All authors contributed to writing the manuscript.

## Funding

## Competing interests
The authors declare no competing interests.
