## [Peer Review File · Nature Communications]

Understanding quantum machine learning also requires rethinking generalizationREVIEWER COMMENTS

Reviewer #1 (Remarks to the Author):

In this manuscript, the authors use numerics (also supported by theoretical results which I think not essential) to show that the most commonly used complexity measures (like the VC dimension, the Rademacher complexity and all their uniform relatives) from traditional statistical learning theory pessimistically estimate the performance of quantum machine learning models. The authors argue that these findings highlight the need for a paradigm shift in the design of quantum models for machine learning. Let me explain their result briefly and roughly/non-accurately in the following.

The performance of a machine learning model = training error + generalization gap. In the traditional learning theory or the complexity measures the author mentioned in this manuscript, both these two terms are properties of models (or more professionally hypothesis class) instead of properties of both model and data. So the training error is pretty small usually means the model has large memory capacity / expressive power if without taking advantage of the structure of data when designing the model. In this case, the generalization gap is usually large. This corresponds to the overfitting regime. In contrast, the generalization gap is pretty small usually means the model has small expressive power or equivalently the model is simple. In this case, the training error is usually large. This corresponds to the underfitting regime. There is a trade-off between these two terms. In order to make both terms small, according to the understanding of traditional statistical learning theory, the model should be both simple enough (i.e., not too many training parameters) and capture the structure of data properly.

However, in this manuscript, they authors show a different picture deviating from traditional statistical learning theory. There are 3 types of data: a. "natural data" (data from phase classification problem on a physical Hamiltonian), b. randomized data from a (which eliminate the physical structure of the data) and c. corrupted data (which can be viewed as a kind of interpolation between a and b). The authors use Quantum Convolutional Neural Network (QCNN) as the model for all these types of data. For b, the authors use both numerics and analytical results to show the training error is very small. Which means QCNN should have large enough memory capacity. Then due to the limit amount of data, we expect this would imply large generalization gap. Then the numerics confirms this point. These suggest that the QCNN model used has large enough size which should causes overfitting according to the above picture given by the statistical learning theory. However, when applied to a by the same model, both training error and generalized gap are small. For c, the training error is also small and the generalization gap behaves as the interpolation between the generalization gaps of an and b. In addition, by definition of those complexity measures mentioned above, the generalization gap from them should be even not less than the generalization gap in b. However, the real world problem should be like a. So this implies that generalization gap from traditional learning theory fails to explain the behavior of quantum machine learning model for "real-world" problem. And this also implies that the structure of data influence the performance a lot even using the same model class.

According to the above line of logic, the authors arrive at the conclusion “These findings expose a fundamental challenge in the conventional understanding of generalization in quantum machine learning and highlight the need for a paradigm shift in the design of quantum models for machine learning tasks.” and “Our numerical results suggest that we must reach further than uniform generalization bounds to fully understand quantum machine learning (QML) models”. I cannot fully agree with the authors on these conclusions. According to the recent development of machine learning (mainly deep learning), it is well-known that the picture provided by the “conventional understanding of generalization” and “uniform generalization bounds” does not work. For example, the Large Language Models (LLM) obviously belong to the over-parameterized regime (the authors clearly know this situation according to their discussion) but the performance is great (here I mixed the discussion of classification/regression and generative model, but the spirit should be conveyed). The conclusion from this manuscript sounds like this phenomena is specific to QML but actually not, it exists in classical machine learning and is well-known. Even rephrased as something like “also in QML”, “highlight the need for a paradigm shift” is a little bit over-claimed. I think not too many people think traditional statistical learning theory can explain modern machine learning. There is even a joke from Internet that “the modern application of statistical learning theory is being exam material for ML students”. Besides, classical machine learning people also know the importance of the structure of data as evidenced that the success of ChatGPT partially due to the quality of data (by data distillation and human annotation).

In summary, I think the result of this manuscript is good to know for anyone working on QML. However, the significance might be over-claimed given the widely known situation in classical machine learning.

Reviewer #2 (Remarks to the Author):

The paper considers the issue of generalization in quantum machine learning. The work is inspired by the influential work "Understanding deep learning requires rethinking generalization" from classical machine learning. The paper shows the limitations of uniform generalization bounds in quantum machine learning with numerical simulations consisting of randomization of the data/labels and analytical discussions.

Overall, the paper is well written and addresses interesting questions in QML. While the paper is a valuable contribution, I would judge the paper to be borderline with respect to Nature Communications. The paper's contribution are rather subtle points on uniform generalization and does not explain why/how QML models could achieve good generalization performance.

More detailed comments:

- Quantum convolutional neural networks are used for the numerical simulations. Can the authors comment on the efficient classical simulability of these networks for a large number of qubits? After each layer, many (half) of the qubits are measured, hence the network is quite shallow and entanglement is limited. What are the classical hardness results for such networks?

- For the numerics, it is unclear if the amount of data samples is too small to guarantee randomness in distribution.

Is it possible that instead of the brute force memorization, the results are caused by the relative ease to find patterns that fit the given labels from the small set of data?

- The theoretical results support the argument but are relatively straightforward. Theorem 1 is analogous to a classical classifier that is defined by a linear combination of the training vectors such as in the support vector machine, regression, etc. For the theorem, the vectors are replaced by density matrices but conceptually it is the same and the linear algebra works out as one expects.

- In the Theorem 1 statement M_y probably shouldn't be called "circuit" but rather "observable", like the authors write in the proof of the theorem.

- "W is well-conditioned per construction". Isn't it per hypothesis?

- Theorem 2 is similar to Theorem 1, however now further assumptions are imposed and a PQC classifier is achieved. These assumptions seem to be quite strict. While it is conceivable that the density matrices can be constructed from parameterized quantum circuits, it is less conceivable that the resulting kernel matrix is well-conditioned. Theorem 3 provides a result on finding well-conditioned matrices, but the connection is not entirely clear.

- In Theorem 3 it is unclear how the σ 's are defined with respect to the original ρ 's. It seems that it is not relevant for this theorem that the σ 's are approximations of the ρ 's. This fact is not used in the proof. If the fact is relevant then the approximation should be defined more properly in the theorem statement.

- The subroutine presented in Theorem 2 for implementing the measurement operator appears to be essentially the LCU technique. Given controlled unitaries, we implement a linear combination of the unitaries. Citations to previous works should be given.

- PQCs are usually considered to be candidate circuits in the near-term quantum computing setting (NISQ). While Theorem 2 allows the classifier to be in the PQC framework, the construction appears not very NISQ friendly. We have additional ancilla qubits, the state preparation of the probability distribution, and the select operation, which require substantial quantum resources.

Reviewer #3 (Remarks to the Author):

We have reviewed with interest the manuscript titled “Understanding quantum machine learning also requires rethinking generalization”. In this work, the authors started with QNNs’ learning with original, label-randomized, and state-randomized datasets with a focus on generalization errors. By combining these numerical results and statistical learning theory, the authors pointed out the imperfections of the commonly used complexity measures for QML models, after which a further theoretical construction made the framework more complete.

We found this work interesting, and helpful for a better understanding of QML. The paper is clearly written and the claims are supported by numerical simulations / theoretical constructions (and opening source the source code/data is appreciated). We just have a few questions/comments for the authors:

1) In Fig 3a, the $gen(f)$ of random labels starts to decrease at $N = 10$ since the memorization capability of the applied QCNNs is limited. As we can imagine, when N becomes very large, this $gen(f)$ will approach 0. We wonder whether the authors can expand the range of the “Training set size N ” to visualize this point?

2) In Eq. (3), the authors used $argmin$ (e.g., when $y = 01$, we try to optimize the model towards outputting the probability $(a, 0, b, c)$), instead of $argmax$ (e.g., when $y = 01$, we try to optimize the model towards outputting the probability $(0, 1, 0, 0)$). Is this choice made for better training performance from some engineering perspective?

Response to Reviewer 1

We would like to thank the reviewer for the time and effort of writing the elaborate and highly valuable report. In order to further clarify our proposal, we have answered the reviewer’s comments and we have proceeded with some improvements in the manuscript, which are detailed individually and point by point.

In this manuscript, the authors use numerics (also supported by theoretical results which I think not essential) to show that the most commonly used complexity measures (like the VC dimension, the Rademacher complexity and all their uniform relatives) from traditional statistical learning theory pessimistically estimate the performance of quantum machine learning models. The authors argue that these findings highlight the need for a paradigm shift in the design of quantum models for machine learning. Let me explain their result briefly and roughly/non-accurately in the following.

Reply: We appreciate Reviewer 1’s thorough review of our manuscript and the valuable suggestions provided, which have resulted in an improvement in the clarity and readability of the text. The theoretical results serve to establish a comprehensive foundation for our numerical findings, extending their applicability beyond near-term quantum devices. To emphasize this, we have clarified the goals of the theoretical results in the initial paragraph of Section C. Additionally, we have heeded the suggestion to modify the phrasing in the abstract from “paradigm shift in the design of quantum models” to “paradigm shift in the study of quantum models”, as also mentioned below by Reviewer 1.

The performance of a machine learning model = training error + generalization gap. In the traditional learning theory or the complexity measures the author mentioned in this manuscript, both these two terms are properties of models (or more professionally hypothesis class) instead of properties of both model and data. So the training error is pretty small usually means the model has large memory capacity / expressive power if without taking advantage of the structure of data when designing the model. In this case, the generalization gap is usually large. This corresponds to the overfitting regime. In contrast, the generalization gap is pretty small usually means the model has small expressive power or equivalently the model is simple. In this case, the training error is usually large. This corresponds to the underfitting regime. There is a trade-off between these two terms. In order to make both terms small, according to the understanding of traditional statistical learning theory, the model should be both simple enough (i.e., not too many training parameters) and capture the structure of data properly. However, in this manuscript, they authors show a different picture deviating from traditional statistical learning theory. There are 3 types of data: a. “natural data” (data from phase classification problem on a physical Hamiltonian), b. randomized data from a (which eliminate the physical structure of the data) and c. corrupted data (which can be viewed as a kind of interpolation between a and b). The authors use Quantum Convolutional Neural Network (QCNN) as the model for all these types of data. For b, the authors use both numerics and analytical results to show the training error is very small. Which means QCNN should have large

enough memory capacity. Then due to the limit amount of data, we expect this would imply large generalization gap. Then the numerics confirms this point. These suggest that the QCNN model used has large enough size which should causes overfitting according to the above picture given by the statistical learning theory. However, when applied to a by the same model, both training error and generalized gap are small. For c, the training error is also small and the generalization gap behaves as the interpolation between the generalization gaps of a and b. In addition, by definition of those complexity measures mentioned above, the generalization gap from them should be even not less than the generalization gap in b. However, the real world problem should be like a. So this implies that generalization gap from traditional learning theory fails to explain the behavior of quantum machine learning model for “real-world” problem. And this also implies that the structure of data influence the performance a lot even using the same model class.

Reply: These are good and valid points. The account given here by Reviewer 1 has made us realize that the terminology used when discussing the mathematics of supervised learning is different from the language sometimes used by ML practitioners. To address this, we have included two additional paragraphs at the end of Section II.A to draw bridges between both languages.

According to the above line of logic, the authors arrive at the conclusion “These findings expose a fundamental challenge in the conventional understanding of generalization in quantum machine learning and highlight the need for a paradigm shift in the design of quantum models for machine learning tasks.” and “Our numerical results suggest that we must reach further than uniform generalization bounds to fully understand quantum machine learning (QML) models.”. I cannot fully agree with the authors on these conclusions. According to the recent development of machine learning (mainly deep learning), it is well-known that the picture provided by the “conventional understanding of generalization” and “uniform generalization bounds” does not work. For example, the *large language models* (LLM) obviously belong to the over-parameterized regime (the authors clearly know this situation according to their discussion) but the performance is great (here I mixed the discussion of classification/regression and generative model, but the spirit should be conveyed). The conclusion from this manuscript sounds like this phenomena is specific to QML but actually not, it exists in classical machine learning and is well-known. Even rephrased as something like “also in QML”, “highlight the need for a paradigm shift” is a little bit over-claimed. I think not too many people think traditional statistical learning theory can explain modern machine learning. There is even a joke from Internet that “the modern application of statistical learning theory is being exam material for ML students”. Besides, classical machine learning people also know the importance of the structure of data as evidenced that the success of ChatGPT partially due to the quality of data (by data distillation and human annotation).

Reply: We acknowledge Reviewer 1’s partial disagreement with our conclusions and have made changes accordingly. We separate the statements and address them individually.

- “These findings expose a fundamental challenge in the conventional understanding of generalization in quantum machine learning and highlight the need for a paradigm shift in the design

of quantum models for machine learning tasks.”

– We have addressed this concern by modifying the statement from “paradigm shift in the design of quantum models” to “paradigm shift in the study of quantum models”. This adjustment aims to clarify our intention, which is to emphasize the importance of designing quantum models with the goal of enhancing our understanding of their generalization capabilities. In particular, we wanted to clarify that the current way generalization is studied, through traditional statistical learning theory, is not the right approach to understanding generalization, and we anticipate that our findings will stimulate further research studying non-uniform approaches for QML models. We recognize that our initial phrasing might be read as an overstatement, as it could be interpreted as proposing specific design principles to improve the overall performance of QML models, which was not our intention.

- “Our numerical results suggest that we must reach further than uniform generalization bounds to fully understand quantum machine learning (QML) models. I cannot fully agree with the authors on these conclusions. According to the recent development of machine learning (mainly deep learning), it is well-known that the picture provided by the ‘conventional understanding of generalization’ and ‘uniform generalization bounds’ does not work.”

– As mentioned by Reviewer 1, we know that uniform approaches to generalization must fail for large, overparametrized, classical deep neural networks. Indeed, LLMs are a good example, there is no fundamental difference in this case between generative and supervised learning. However, it is imperative to recognize that classical and quantum models are fundamentally different. They not only operate on distinct computing platforms of a physically different nature, but also, the functions produced by neural networks are typically different from those produced by parametrized quantum circuits. As a consequence, caution is warranted in expecting these two different learning models to behave equally when faced with randomization experiments based on unrelated learning tasks. The fact that the quantum and classical learning models display similar results should not be taken for granted, especially for small quantum models. We have elaborated on this discussion in Section III for further clarity.

- We take special notice of Reviewer 1’s statement: “The conclusion from this manuscript sounds like this phenomena is specific to QML but actually not, it exists in classical machine learning and is well-known.”

– Here, we would take the liberty to politely disagreeing with Reviewer 1, as this is the opposite of the message we wanted to convey. But of course, we see the point that we should have explained this point better. We have carefully reconsidered the discussion in Section III to ensure clarity regarding the observed phenomenon. At no point do we mention this phenomenon is specific to QML. In fact, we acknowledge the existence of analogous phenomena in classical machine learning by directly mentioning the paper that initiated this line of thought from deep learning [1], which we have cited multiple times both in the previous and new versions of the manuscript. The title of our manuscript is a clear wink to this foundational

reference, reflecting our awareness of the broader literature. We believe the revisions in Section III effectively address this concern, but we remain open to further suggestions from Reviewer 1 to improve the clarity of our message.

In summary, I think the result of this manuscript is good to know for anyone working on QML. However, the significance might be over-claimed given the widely known situation in classical machine learning.

Reply: We appreciate Reviewer 1’s valuable feedback on our manuscript and the overall positive response. The suggested revisions have enhanced the clarity and coherence of our message, in our assessment. Regarding the significance of our work, we respectfully assert that our results are more than “good to know for anyone working on QML”. After careful consideration of the existing literature in QML, we honestly think that our manuscript is actually “important for anyone working on QML”, and at least “good to know for potential applications of QML”, including quantum computing-based approaches to classifying ground states of many-body Hamiltonians and other learning tasks about physical properties from data.

We suspect that there may be a difference of opinion between Reviewer 1 and ourselves regarding “what was already known in the field of generalization in QML.” In our response, we have elucidated the relation between our manuscript and the seminal Ref. [1]. We summarize our perspective on the state of knowledge before the emergence of our results:

- We believe researchers in QML were aware of the classical literature and the modern approaches to generalization for large overparametrized neural networks.
- The main message of Ref. [1] is that traditional learning theory fails to explain the success of *deep convolutional neural networks*.
- On the one hand, we discuss the difference between large classical models and current quantum models in terms of scale (discussed in Section III).
- On the other hand, there is a point to be made on the distinction between *classical neural networks* and *quantum neural networks*. Recent efforts have been directed towards understanding the comparative behavior of classical and quantum models. It is not directly apparent that quantum neural networks should display the same behavior as classical neural networks when it comes to exhibiting good generalization even in the case of overfitting. Regarding randomization tests in particular, these are specific to different model-task configurations (also now discussed in Section III).
- The research in generalization in QML over the past few years has seen numerous papers employing traditional learning theory. In fact, works in this vein started emerging in 2020 [2], well after Ref. [1]. We cite some of these studies [2, 3, 4, 5, 6, 7, 8, 9, 10, 11, 12, 13, 14, 15, 16, 17] as evidence of the uncertainty surrounding the direct applicability of results from classical literature to QML, especially within near-term QML.

Finally, in a broader context, our findings hold deep implications for all works studying traditional learning theory applied to QML. We do not imply that those results are false, but instead, we show they are vacuous in practice. In particular, this extends to recent publications in *Nature Communications* [11, 12]. Again, we would like to thank the reviewer for the elaborate report. The constructive criticism expressed there has helped us to substantially sharpen our core message. Having accommodated the concerns, we hope that our work is now suitable for publication in its present form.

Response to Reviewer 2

We would like to thank the reviewer wholeheartedly for the elaborate and helpful report, and for the time invested in reviewing the previous version of our manuscript.

The paper considers the issue of generalization in quantum machine learning. The work is inspired by the influential work “Understanding deep learning requires rethinking generalization” from classical machine learning. The paper shows the limitations of uniform generalization bounds in quantum machine learning with numerical simulations consisting of randomization of the data/labels and analytical discussions.

Reply: Thanks again, we believe Reviewer 2 has clearly understood the overall message and ideas of our manuscript.

Overall, the paper is well written and addresses interesting questions in QML. While the paper is a valuable contribution, I would judge the paper to be borderline with respect to *Nature Communications*. The paper’s contribution are rather subtle points on uniform generalization and does not explain why/how QML models could achieve good generalization performance.

Reply: We appreciate Reviewer 2’s positive assessment of our manuscript. Of course, we are delighted to read that our work is “well written and addresses interesting questions in QML”. Regarding the significance of our manuscript in a broader context, we are now convinced that our presentation of a few key ideas in the first version could have been clearer. It is well possible that we have been a bit carried away by our results and have not elaborated sufficiently on the broader implications of our work. We have addressed this problem by heavily re-working the discussion in Section III.

We would like to point out that regarding the question of “why/how QML models could achieve good generalization performance”, it is important to note that this remains a challenging question even in the mature field of classical machine learning, and in fact, it is the final goal of the entire field regarding generalization. We believe it would be an unattainable expectation for a paper in QML. The literature includes many proposals studying restricted cases of uniform explanations for generalization in QML, some of which we have cited [2, 3, 4, 5, 6, 7, 8, 9, 10, 11, 12, 13, 14, 15, 16, 17].

Our contribution demonstrates that these proposals are already vacuous for existing quantum models, including those published in *Nature Communications* [11, 12]. While we do not provide a complete solution to the field, our work problematizes essentially the entire literature on generalization in QML. We anticipate that our findings will stimulate further research studying non-uniform approaches. We believe this represents a significant impact in the field of QML, with broad implications for applications such as classifying phases of ground states of many-body Hamiltonians.

More detailed comments: - Quantum convolutional neural networks are used for the numerical simulations. Can the authors comment on the efficient classical simulability of these networks for

a large number of qubits? After each layer, many (half) of the qubits are measured, hence the network is quite shallow and entanglement is limited. What are the classical hardness results for such networks?

Reply: We have included a paragraph in Section II.B.1 explaining that the QCNN cannot be simulated for arbitrary input states. The logarithmic circuit depth makes the circuit simulable for, e.g., low-entanglement input states that admit efficient MPS representations. However, if the input state cannot be efficiently represented and stored in classical memory, classical simulation of the circuit becomes infeasible.

For the numerics, it is unclear if the amount of data samples is too small to guarantee randomness in distribution. Is it possible that instead of the brute force memorization, the results are caused by the relative ease to find patterns that fit the given labels from the small set of data?

Reply: We appreciate Reviewer 2’s inquiry regarding the potential influence of data sample size on the observed results. This is a key idea that we want to make sure is clear upon first reading of the manuscript. In response to Reviewer 2’s query, our reworking of the discussion in Section III also aims at addressing it. We emphasize that our use of the term memorization refers to the capability of a model to perfectly fit the training data while failing to generalize to unseen data. This concept is now elaborated upon in Section II.A. With this standard definition, “the relative ease to find patterns that fit a small training set” is just a manifestation of the fact that it is easier to memorize a small training set. For instance, if the training set consisted of only one point, that would be considered memorization.

We acknowledge the importance of this distinction, as it underlies our experimental design. Our randomization experiments demonstrate that the QCNN can memorize training sets of sizes up to 20. Notably, the same QCNN, with the same number of training examples, learns (not memorizes) the true data distribution, achieving comparable training accuracy to the randomized cases. This distinction forms the basis for our problem size and training set selection. We are aware, also from Ref. [11], that the QCNN is capable of *learning* this phase diagram with these many training points. We show that it is also capable of *memorizing* sets of the same size, thus leading to the failure of uniform generalization bounds. We understand this message needed to be clearer in the previous version. We hope our additions to Sections II.A and III have clarified these points, addressing any prior ambiguity.

The theoretical results support the argument but are relatively straightforward. Theorem 1 is analogous to a classical classifier that is defined by a linear combination of the training vectors such as in the support vector machine, regression, etc. For the theorem, the vectors are replaced by density matrices but conceptually it is the same and the linear algebra works out as one expects.

Reply: We agree with the assessment of Reviewer 2, although it is to an extent unclear to us whether Reviewer 2’s remark should be taken as criticism, as we see no clear actionable advice from it.

Initially, we considered skipping Theorem 1 and directly presenting Theorem 2, which would provide limited insight into the proof of the latter in the main text. However, we decided to include Theorem 1 to facilitate the reader’s comprehension of the underlying principles leading to Theorem 2.

As Reviewer 2 notes, the construction of Theorem 1 is an application of linear regression. The construction of Theorem 2 builds upon the concepts established in Theorem 1, with the additional constraint that part of the calculation can be written as a PQC. This approach was taken to elucidate the mathematical underpinnings for the benefit of the reader. At the same time, we would argue that the difficulty of the proof of a theorem does not affect the validity of the statement.

In the Theorem 1 statement M_y probably shouldn’t be called “circuit” but rather “observable”, like the authors write in the proof of the theorem.

Reply: We have modified Theorems 1 and 2 as advised.

“W is well-conditioned per construction”. Isn’t it per hypothesis?

Reply: We replaced “construction” by “hypothesis” in Appendix Section A.

Theorem 2 is similar to Theorem 1, however now further assumptions are imposed and a PQC classifier is achieved. These assumptions seem to be quite strict. While it is conceivable that the density matrices can be constructed from parameterized quantum circuits, it is less conceivable that the resulting kernel matrix is well-conditioned. Theorem 3 provides a result on finding well-conditioned matrices, but the connection is not entirely clear.

Reply: We agree with Reviewer 2 that the applicability of Theorem 2 was unclear in the previous version of the manuscript. Our goal was only to establish sufficient conditions and show *how* PQCs of arbitrary size could be universal in the sense of finite sample expressivity rather than proving universality in all cases. To clarify this point, we have included a sentence in the first paragraph of Section II.C.

Furthermore, we also agree with Reviewer 2 that the connection between Theorems 2 and 3 has not been sufficiently clearly stated in the previous version of the manuscript. In response, we have included an explanatory paragraph following the proof of Theorem 3, explicitly stating the relation between these two theorems. Lastly, it is important to note that \hat{W} may not always be a kernel matrix, as the approximations $\hat{\rho}_s$ of ρ_s may not necessarily be accurate, as we elaborate in the following reply.

In Theorem 3 it is unclear how the sigma’s are defined with respect to the original rho’s. It seems that it is not relevant for this theorem that the sigma’s are approximations of the rho’s. This fact is not used in the proof. If the fact is relevant then the approximation should be defined more properly in the theorem statement.

Reply: We have addressed the concerns raised by Reviewer 2 regarding the relation between ρ and σ in Theorem 3. In particular, we have included an additional paragraph after Algorithm 1 to clarify that the success of the algorithm does not strictly depend on the quality of the approximation of σ s to ρ s. While it is indeed a sufficient condition, provided the Gram matrix W is well-conditioned, it is not a necessary one for the algorithm to succeed. The key criterion is the well-conditioning of the matrix of inner products \hat{W} , which may hold even if the initial states ρ are poorly approximated. For instance, \hat{W} could end up being proportional to a non-trivial permutation matrix, in which case it would be well-conditioned, and some of the $\hat{\rho}$ s would be far away from the corresponding ρ s.

We have tried to phrase the results with the goal of not being more restrictive than necessary, but we recognize that the previous version may have been overly abstract in its formulation.

The subroutine presented in Theorem 2 for implementing the measurement operator appears to be essentially the LCU technique. Given controlled unitaries, we implement a linear combination of the unitaries. Citations to previous works should be given.

Reply: We thank Reviewer 2 for pointing out to us this relevant prior literature. We have added a reference after Theorem 2.

PQCs are usually considered to be candidate circuits in the near-term quantum computing setting (NISQ). While Theorem 2 allows the classifier to be in the PQC framework, the construction appears not very NISQ friendly. We have additional ancilla qubits, the state preparation of the probability distribution, and the select operation, which require substantial quantum resources.

Reply: As noted by Reviewer 2, PQCs are commonly considered within the NISQ context. Our randomization experiments are conducted within this framework, where we identify good generalization behavior of the QCNN architecture. Having shown the limitations of uniform generalization bounds for near-term PQCs, our goal is to show that our results also carry over beyond the near-term regime. This is precisely the motivation behind our analytical results. We have addressed this clarification in the first paragraph of Section II.C.

We would like to once again thank Reviewer 2 for their thoughtful and valuable comments. Their suggestions have enabled us to present our ideas in a substantially more organized and clear manner. Having accommodated all comments and concerns, we hope that our work is now suitable for publication in its present form.

Reponse to Reviewer 3

We would like to thank Reviewer 3 for the time invested in reviewing and for their kind assessment of the manuscript. The report has been very helpful for us.

We have reviewed with interest the manuscript titled “Understanding quantum machine learning also requires rethinking generalization”. In this work, the authors started with QNNs’ learning with original, label-randomized, and state-randomized datasets with a focus on generalization errors. By combining these numerical results and statistical learning theory, the authors pointed out the imperfections of the commonly used complexity measures for QML models, after which a further theoretical construction made the framework more complete. We found this work interesting, and helpful for a better understanding of QML. The paper is clearly written and the claims are supported by numerical simulations / theoretical constructions (and opening source the source code/data is appreciated).

Reply: We thank the reviewer for this positive assessment of our work. Needless to say, we are delighted to see that the reviewer “found this work interesting, and helpful for a better understanding of QML”. We also share the sentiment on code and data availability.

We just have a few questions/comments for the authors:

In Fig 3a, the $\text{gen}(f)$ of random labels starts to decrease at $N = 10$ since the memorization capability of the applied QCNNs is limited. As we can imagine, when N becomes very large, this $\text{gen}(f)$ will approach 0. We wonder whether the authors can expand the range of the “Training set size N ” to visualize this point?

Reply: We appreciate Reviewer 3’s valuable suggestion regarding expanding the range of the training set size to visualize how the generalization gap $\text{gen}(f)$ for real and random labels will approach zero. While we agree that exploring the behavior of $\text{gen}(f)$ as N tends to larger values would show a nice curiosity, we believe that in the context of our current study, it will not add substantial value to the results we aim to convey. Our primary focus lies within a practical and relevant range of training set sizes. Expanding the range to larger N introduces additional usage of heavy computational resources without yielding significant insights to our central findings. Therefore, we have decided to maintain the current range of sizes.

In Eq. (3), the authors used argmin (e.g., when $y = 01$, we try to optimize the model towards outputting the probability $(a, 0, b, c)$), instead of argmax (e.g., when $y = 01$, we try to optimize the model towards outputting the probability $(0, 1, 0, 0)$). Is this choice made for better training performance from some engineering perspective?

Reply: This classification rule and loss function, which involve selecting the outcome with the *lowest* probability, was already utilized in Ref. [11]. The authors found that employing this seemingly counter-intuitive loss function could lead to good generalization performance. In our manuscript, we

reproduced their experimental design. We have added a paragraph in Section II.B.1 for clarification.

Again, we thank the reviewer for the time invested in the report and, at the same time, the positive response. It has been helpful for us. Having accommodated the comments, we hope that our work is now suitable for publication in its present form.

References

- [1] C. Zhang, S. Bengio, M. Hardt, B. Recht, and O. Vinyals. Understanding deep learning requires rethinking generalization. In *Int. Conf. Learn. Rep.*, 2017.
- [2] M. C. Caro and I. Datta. Pseudo-dimension of quantum circuits. *Quant, Mach. Intell.*, 2:14, 2020.
- [3] A. Abbas, D. Sutter, C. Zoufal, A. Lucchi, A. Figalli, and S. Woerner. The power of quantum neural networks. *Nature Comp. Sc.*, 1(6):403–409, 2021.
- [4] L. Banchi, J. Pereira, and S. Pirandola. Generalization in quantum machine learning: A quantum information standpoint. *PRX Quantum*, 2(4):040321, 2021.
- [5] K. Bu, D. E. Koh, L. Li, Q. Luo, and Y. Zhang. Effects of quantum resources and noise on the statistical complexity of quantum circuits. *Quant. Sc. Tech.*, 8(2):025013, 2023.
- [6] K. Bu, D. E. Koh, L. Li, Q. Luo, and Y. Zhang. Rademacher complexity of noisy quantum circuits. *arXiv:2103.03139*, 2021.
- [7] K. Bu, D. E. Koh, L. Li, Q. Luo, and Y. Zhang. Statistical complexity of quantum circuits. *Phys. Rev. A*, 105(6):062431, 2022.
- [8] Y. Du, Z. Tu, X. Yuan, and D. Tao. Efficient measure for the expressivity of variational quantum algorithms. *Phys. Rev. Lett.*, 128(8):080506, 2022.
- [9] C. Gyurik and V. Dunjko. Structural risk minimization for quantum linear classifiers. *Quantum*, 7:893, 2023.
- [10] M. C. Caro, E. Gil-Fuster, J. Jakob Meyer, J. Eisert, and R. Sweke. Encoding-dependent generalization bounds for parametrized quantum circuits. *Quantum*, 5:582, November 2021.
- [11] M. C. Caro, H.-Y. Huang, M. Cerezo, K. Sharma, A. Sornborger, L. Cincio, and P. J. Coles. Generalization in quantum machine learning from few training data. *Nature Comm.*, 13(1):4919, 2022.
- [12] M. C. Caro, H.-Y. Huang, N. Ezzell, J. Gibbs, A. T. Sornborger, L. Cincio, P. J. Coles, and Z. Holmes. Out-of-distribution generalization for learning quantum dynamics. *Nature Comm.*, 14:3751, 2023.
- [13] Y. Qian, X. Wang, Y. Du, X. Wu, and D. Tao. The dilemma of quantum neural networks. *IEEE Trans. Neu. Net. Learn. Sys.*, pages 1–13, 2022.
- [14] Y. Du, Y. Yang, D. Tao, and M.-H. Hsieh. Demystify problem-dependent power of quantum neural networks on multi-class classification. *arXiv:2301.01597*, 2022.

- [15] L. Schatzki, M. Larocca, F. Sauvage, and M. Cerezo. Theoretical guarantees for permutation-equivariant quantum neural networks. *arXiv:2210.09974*, 2022.
- [16] E. Peters and M. Schuld. Generalization despite overfitting in quantum machine learning models. *arXiv:2209.05523*, 2022.
- [17] T. Haug and M. S. Kim. Generalization with quantum geometry for learning unitaries. *arXiv:2303.13462*, 2023.

REVIEWERS' COMMENTS

Reviewer #1 (Remarks to the Author):

If the authors claim the main result as "it is not directly apparent that the quantum neural networks should display the same behavior as classical neural networks when it comes to exhibiting good generalization even in the case of overfitting" as commented in their reply and this study this in detail at least for some examples like randomized data, then I agree with Reviewer 2 that this paper is on the borderline of Nature Communications.

Although I still think claiming the limitation of traditional learning theory is not a big deal (which can be verified by talking to most of the modern machine learning practitioners), according to the authors' reply that there are already two papers on traditional learning theory for QML published on Nature Communications, this manuscript should worth being published.

Reviewer #2 (Remarks to the Author):

Regarding Theorem 3:

- The first "then" of the theorem in "Then, the real numbers" seems clearly not necessary. This sentence is defining quantities.
- Isn't the theorem also assuming that we have been given the $\text{tr}\{\rho_i \sigma^j_k\}$?
- Polynomial time when we have the traces. Without traces also poly time?
- After the theorem: "it suffices for us to assume that the resulting matrix W^{\wedge} is well-conditioned". Using the word "assume" here seems incorrect. This is a result of the theorem, something the theorem can certify. See the third "then" of the theorem statement, "one can then decide in polynomial time whether".

Generally the paper should be interesting to the community.

Reviewer #3 (Remarks to the Author):

The authors have addressed my comments with clarity and in detail. With the revised version of the manuscript, I'm happy to recommend it for publication in Nature Communications.

Response to Reviewer 2

We sincerely thank the reviewer for the thoughtful report.

Regarding Theorem 3:

- The first "then" of the theorem in "Then, the real numbers" seems clearly not necessary. This sentence is defining quantities.

Reply: We thank the referee for the pointer. We have amended it.

- Isn't the theorem also assuming that we have been given the $\text{tr}(\rho_i \sigma_k^j)$?

Reply: We agree with the referee. We now explicitly spell it out for completeness.

- Polynomial time when we have the traces. Without traces also poly time?

Reply: We had implicitly assumed this, and now we have added it explicitly for completeness. We thank the referee for pointing this out, as now it should be clear for everyone who reads the theorem.

- After the theorem: "it suffices for us to assume that the resulting matrix \hat{W} is well-conditioned". Using the word "assume" here seems incorrect. This is a result of the theorem, something the theorem can certify. See the third "then" of the theorem statement, "one can then decide in polynomial time whether".

Reply: We completely agree with the referee. We have amended the sentence.

Generally the paper should be interesting to the community.

We want to thank Referee 2 again for the time invested in the report and for their thoughtful comments.